# The Effects of Outdoor versus Indoor Exercise on Psychological Health, Physical Health, and Physical Activity Behaviour: A Systematic Review of Longitudinal Trials

**DOI:** 10.3390/ijerph20031669

**Published:** 2023-01-17

**Authors:** Matt Noseworthy, Luke Peddie, E. Jean Buckler, Faith Park, Margaret Pham, Spencer Pratt, Arpreet Singh, Eli Puterman, Teresa Liu-Ambrose

**Affiliations:** 1Aging, Mobility, and Cognitive Health Lab, University of British Columbia, Vancouver, BC V6T 2B5, Canada; 2School of Kinesiology, University of British Columbia, Vancouver, BC V6T 1Z1, Canada; 3School of Exercise Science, Physical and Health Education, University of Victoria, Victoria, BC V8W 2Y2, Canada

**Keywords:** green exercise, outdoor exercise, nature, physical health, psychological health, physical activity behaviour

## Abstract

A growing body of research is exploring the potential added health benefits of exercise when performed outdoors in nature versus indoors. This systematic review aimed to compare the effects of exercise in outdoor environments versus indoor environments on psychological health, physical health, and physical activity behaviour. We searched nine databases from inception to March 2021 for English language, peer-reviewed articles: MEDLINE, Embase, PubMed, Scopus, Web of Science, CINAHL, SportsDiscus, GreenFile, and CENTRAL. We included randomized and non-randomized trials that compared multiple bouts of exercise in outdoor versus indoor environments, and that assessed at least one outcome related to physical health, psychological health, or physical activity behaviour. Due to minimal outcome overlap and a paucity of studies, we performed a narrative synthesis. We identified 10 eligible trials, including 7 randomized controlled trials, and a total of 343 participants. Participant demographics, exercise protocols, and outcomes varied widely. In the 10 eligible studies, a total of 99 comparisons were made between outdoor and indoor exercise; all 25 statistically significant comparisons favoured outdoor exercise. Interpretation of findings was hindered by an overall high risk of bias, unclear reporting, and high outcome heterogeneity. There is limited evidence for added health or behaviour benefits of outdoor exercise versus indoor exercise. Rigorous randomized controlled trials are needed with larger samples and clear reporting.

## 1. Introduction

Physical activity has a multitude of benefits for physical and mental health. For instance, robust observational and experimental evidence indicates that regular physical activity or exercise can help prevent or manage multiple chronic physical conditions, such as cardiovascular disease, metabolic syndrome, and cancer, as well as chronic psychiatric conditions, such as depression and anxiety [1,2,3,4]. Despite these benefits, globally, more than 25% of adults and more than 80% of adolescents do not meet recommended minimum levels of physical activity [1,5]. More research is needed to determine effective strategies for promoting physical activity participation and to clarify optimal exercise parameters for protective effects [1,2,6,7].

One factor that may modulate the benefits and behaviours associated with physical activity is the environment in which it is performed—specifically, the extent of natural versus built (e.g., streets, buildings) environmental features. The term “green exercise” [8] was introduced in 2003 as an umbrella term to describe physical activity in the presence of nature, ranging from full immersion in outdoor, natural environments to indoor exposures to nature elements (e.g., views, images, smells, sounds). Emerging experimental evidence suggests that exercise performed in outdoor, natural environments (i.e., outdoor green exercise) may confer additional health benefits, as compared with exercise in outdoor, built environments [9,10,11,12,13] or indoor environments [14,15,16].

Given the rise in green exercise research over the past decade (Figure A1), two systematic reviews have been conducted to examine the effects of outdoor green exercise versus indoor exercise on physical and mental health [15,16]. In 2011, Thompson Coon et al. [15] systematically reviewed studies of any design that compared physical and mental well-being outcomes between outdoor exercise and indoor exercise. The authors identified 11 eligible studies, all of which were single-bout studies assessing mainly mental outcomes in healthy adults. A narrative synthesis highlighted the favourable effects of outdoor exercise for several psychological outcomes (e.g., positive and negative emotions, energy, and enjoyment) [15]. Evidence was limited by a dearth of studies, small sample sizes, high outcome heterogeneity, and poor methodological quality [15].

In 2019, Lahart et al. [16] systematically reviewed experimental or quasi-experimental studies comparing exercise in outdoor, natural environments or in the presence of simulated nature (e.g., images or videos) versus indoor environments without nature exposure. They identified three longitudinal trials and 25 single-bout trials. Overall, while showing the emergence of longitudinal trials and an increase in physical outcomes since the 2011 review, the findings of the 2019 review were otherwise fairly similar: inconclusive evidence for the benefits of an outdoor, natural exercise environment, a high risk of bias across studies, and a need for more rigorous designs and more longitudinal trials [16].

Since 2019, the increase in green exercise studies has continued (Figure A1), including an increase in longitudinal green exercise studies, e.g., [17,18,19,20]. Additionally, while this growing body of research suggests that outdoor green exercise may have greater health benefits than exercise performed in built environments, previous green exercise reviews [15,16] have not explored (1) the potential moderating effect of the extent of natural versus built elements in the outdoor environment, and (2) the relative effects of outdoor versus indoor built environments. By comparing exercise in any outdoor environment—regardless of the extent of natural elements—to indoor exercise, we planned to assess the quality of the environment (i.e., natural, built, or mixed) as a potential moderator of the effects of outdoor exercise. Our systematic review aimed to update and extend the prior systematic reviews, by synthesizing longitudinal studies comparing the effects of outdoor exercise versus indoor exercise on psychological health, physical health, and physical activity behaviour.

## 2. Materials and Methods

### 2.1. Review Protocol

This systematic review followed the PRISMA guidelines for reporting systematic reviews [21] (see Appendix A for the checklist). The study protocol was originally registered on PROSPERO on 24 July 2018 under registration number, CRD42018100314. However, during the initial article screening, we decided to divide the study into distinct reviews for acute (i.e., one exercise bout per environmental condition) and longitudinal (i.e., two or more exercise bouts per environmental condition) interventions. The original protocol (CRD42018100314) was updated to focus on acute trials, while a new protocol (CRD42020180756) was registered on 14 July 2020 to focus on longitudinal trials.

### 2.2. Search Strategy and Screening

The following nine databases were searched for English language, peer-reviewed articles from inception to 23 March 2021: MEDLINE, Embase, PubMed, Scopus, Web of Science, CINAHL, SportsDiscus, GreenFile, and CENTRAL. Search terms were selected by members of the research team, in consultation with the subject librarian and previous systematic reviews of relevant topics [15]. As seen in the master search strategy (Table 1), four major categories were used to search titles and abstracts: exercise; outdoors; outdoor exercise; and indoors. The full search strategy for each database is presented in Appendix A. Previous systematic reviews and reference lists of included articles were hand-searched. The database search was originally conducted on 9 July 2018, but we updated the search in November 2019 and in March 2021; we used the same search protocol, except that we limited the time window to the date of the previous search onwards.

One author (L.P., F.P., or M.N.) conducted the searches and removed duplicates. At least two authors (L.P., E.J.B., A.S., F.P., or M.N.) independently screened titles, then abstracts, for eligibility. At least two authors (L.P., E.J.B., S.P., A.S., F.P., or M.N.) independently reviewed full-text articles for eligibility. After each stage, authors decided on a consensus for any discrepancies by discussion and, if necessary, by consulting with a senior author (E.P., T.L.A.).

### 2.3. Eligibility Criteria

Studies were screened according to the following eligibility criteria, as summarized in Table 2: at least one main outcome related to physical or psychological health or physical activity behaviour (e.g., adherence or intention to participate again), an experimental design comparing an outdoor exercise intervention to an indoor exercise control, and at least two bouts of exercise for each environmental condition. No limits were placed on exercise type, duration, or intensity, so long as the outdoor and indoor conditions followed the same or sufficiently similar exercise protocols. To ensure the inclusion of all relevant studies and to fully explore the effects of the spectrum of outdoor environments, the current review included all studies that described the exercise environment as outdoors, regardless of the extent of natural elements. We excluded studies in which the outdoor condition was substituted with an indoor simulation of an outdoor environment (e.g., virtual reality, projection, sensory stimulation), or in which the indoor condition involved explicit exposure to natural elements or simulations of an outdoor environment.

### 2.4. Data Extraction

Data extraction and risk of bias assessment were documented using Covidence, an online systematic review management tool [22]. Two authors (F.P., M.P., M.N., or L.P.) independently extracted data from articles meeting eligibility criteria. Data were extracted on (1) general study information, such as country and setting in which the research took place, the theoretical framework under which the study was conducted, and the study purpose; (2) study methodology, including study design, eligible sample size, participant characteristics, exercise protocols, description of outdoor and indoor environments, and study duration; and (3) eligible outcomes and results, including any outcomes related to psychological or physical health or physical activity behaviour, measurement tools, statistical analysis, and effects on each outcome (groupwise means, standard deviations, sample sizes, effect sizes if available, and statistically significant effects as reported in each study). Sample size and participant characteristics were sought only for the relevant outdoor and indoor exercise groups in studies that involved additional treatment and/or control groups. Two authors (M.N., L.P.) compared data extraction forms and reviewed relevant articles for correction in the case of any discrepancies. Our primary effect measure for eligible outcomes was between-group differences in post-intervention group means (preferably while controlling for baseline group means) or in change scores (from baseline to post-intervention).

### 2.5. Risk of Bias Assessment

Two authors (F.P., M.P., M.N., or L.P.) independently assessed the risk of bias for each of the included studies using the Revised Cochrane risk of bias (RoB) tool for randomized trials (RoB-2) [23]. For all parallel-group trials, we used the RoB 2 version designed for individually randomized parallel-group trial designs, assessing the effect of assignment to interventions, i.e., “intention-to-treat”. For one cluster-randomized, crossover trial [17], we used the RoB 2 tool supplements for cluster-randomized trials and for crossover trials.

The RoB 2 tool categorizes the risk of bias into five domains, with an additional domain for crossover trials: (1) bias arising from the randomization process; (2) bias due to deviations from intended interventions; (3) bias due to missing outcome data; (4) bias in measurement of the outcome; (5) bias in selection of the reported result; and (S) bias arising from period and carryover effects [23]. In particular, for Domain 5, we assessed selective outcome reporting bias by comparing the reported outcomes and analyses with those prespecified in trial protocols, if available, or in the methods sections of the trial publications.

The risk of bias assessment proceeded in three stages. First, each reviewer answered a series of signalling questions within each domain. Secondly, each reviewer followed an algorithm to judge the level of risk of bias (low, some concerns, or high) for each domain based on responses to signalling questions. Finally, any discrepancies in the risk of bias judgements were resolved by discussion between at least two reviewers to determine a consensus judgement. For each study, the overall risk of bias judgement was the highest judgement made in at least one domain, or a high risk of bias if a study was judged to have “some concerns” for multiple domains in such a way as to significantly decrease confidence in the study’s results [23]. To summarize the risk of bias across domains and studies, we produced a bar chart and traffic light plot using the R package “robvis” [24]. 

### 2.6. Data Synthesis

We tabulated individual study results for each eligible outcome, including groupwise sample sizes, means, and standard deviations, as well as effect sizes and statistical results. We had planned to complete a meta-analysis if outcomes were sufficiently homogeneous, and to perform subgroup analysis (on participant characteristics, quality of the outdoor environment, and exercise modality and intensity) if the evidence was available. However, due to the heterogeneity of outcome measures and the paucity of eligible studies, we decided that neither meta-analysis nor subgroup analysis was appropriate. Instead, we performed a narrative synthesis, with outcomes grouped into the three domains of psychological health, physical health, or physical activity behaviour. To help summarize the data, we performed vote counting based on the direction (i.e., favouring outdoor or indoor exercise) and the statistical significance of effects reported in each study. We also prepared a summary table wherein we grouped discrete outcomes into broader outcome categories, and presented the number and direction of significant group differences for each outcome category.

## 3. Results

### 3.1. Search Results

Our electronic searches identified 57144 initial results. We identified an additional 92 articles from other sources, including reviews by Thompson Coon et al. [15] and Lahart et al. [16], as well as hand searches of reference lists of included articles. 35,738 articles remained after duplicate removal. Following title and abstract screening, the remaining 127 articles were divided into acute and longitudinal studies for separate reviews. In the present review of longitudinal studies, following full-text screening, we included 10 trials from 12 articles (with 2 trials having 2 articles each) for narrative synthesis (Figure 1). Three of these longitudinal trials (corresponding to five articles) were included in the 2019 review by Lahart et al. [16].

During title and abstract screening, common topics of irrelevant articles included Parkinson’s disease (likely due to the use of the search term “park*”), comparisons of multiple outdoor exercise conditions with no indoor control (e.g., natural versus urban), sensors for tracking and/or classifying activity and behaviour, and observational studies relating ecological factors with physical activity and/or health.

Reasons for exclusions during full-text screening are listed in Figure 1. We highlight two of these excluded articles, which reported on the same trial, but with different outcomes; this trial compared groups of students receiving different proportions of outdoor and indoor physical education classes [25,26]. We deemed this trial ineligible because it did not compare discrete treatments of outdoor versus indoor exercise.

### 3.2. Study Characteristics

#### 3.2.1. Trial Design Characteristics

Characteristics of included studies are presented in Appendix A. Six of the ten included trials were parallel-group randomized controlled trials [18,20,27,28,29,30,31,32], whereas three studies were parallel-group controlled trials with no reporting of randomization [19,33,34]. One study was a cluster-randomized, counterbalanced crossover trial [17].

All studies aimed to compare the effect of outdoor exercise versus indoor exercise on psychological, physical, and/or behavioural outcomes. Two of these studies focused on specific environmental variables: cold exposure (outdoors at −5°C–5°C versus indoors at 21°C–25°C) [19] and terrain (outdoor multisurface path versus indoor solid floor) [20]. Three of the studies also investigated additional independent variables (i.e., vitamin D supplementation [29] and exercise [18,32]) by including additional intervention groups or control groups [18,29,32] that were not relevant to this review and will not be further discussed.

#### 3.2.2. Participant Characteristics

Sample sizes ranged from 14 [27,28] to 87 [32], with a total sample of 343 participants across the 10 trials (mean ± standard deviation: 34.3 ± 25.5; median: 26.5). These sample sizes exclude additional treatment groups and/or control groups beyond the outdoor and indoor exercise groups relevant to this review. Only three studies reported performing a power calculation to determine sample size [17,18,27]; one other study reported sample size consideration based on a prior study [30,31].

Participant populations varied across all studies (see Appendix A for details). Biological sex was reported in eight trials, including two trials with an approximately even mix of males and females [17,27,28], two trials with mostly females [20,32], and two trials each with all females [29,30,31] or all males [18,19]. One of these trials [32] involved additional control groups (beyond the outdoor and indoor exercise groups), but reported sex only for the parent sample. Mean ages ranged from 11 to 80 years. Half of the trials had a mean participant age under 25 years [17,18,19,32,33]. Two trials involving additional control groups reported mean age only for the parent sample [29,32].

#### 3.2.3. Exercise Environment: Physical Setting and Conditions

We classified the outdoor exercise environments as mostly natural with minimal built (i.e., human-made) features, mostly built, or a mix of natural and built features.

Among the outdoor exercise environments of the 10 included studies, six were not reported in adequate detail (described only as “outdoors”, “open”, or “large park”) [17,19,32,33,34], two were a mix of natural and built features (i.e., turf field; multisurface garden path) [18,20], and two were mostly natural with minimal built features (i.e., forest; riverside park) [27,28,30,31]. The indoor exercise environments consisted of traditional indoor exercise spaces in three studies (i.e., gym, health club) [18,27,28,29], laboratory spaces in one study [30,31], university tunnels/skyways in one study [17], and a nursing home hall in one study [20]. Indoor environments were inadequately reported in four studies (described only as “indoors”, “closed”, or “into the hall”) [19,32,33,34].

Environmental conditions were reported for three trials, including temperature [18,19,27,28], humidity [18,27,28], weather [18,27,28], and altitude [18]. 

#### 3.2.4. Exercise Protocols

We summarized exercise protocols according to exercise time (i.e., duration and frequency), type (e.g., running, cycling, strength), modality (e.g., overground versus treadmill), and intensity (e.g., prescribed intensity range versus self-selected).

##### Time Spent Exercising 

Trial durations ranged from 18 weeks in one study [19] to less than one week per environment for two studies [17,27,28]. Exercise session duration was reported in nine studies, and ranged from 25–60 min. Session frequency was reported in nine studies and ranged from one to five sessions per week (median: three).

Taking together the trial duration, session duration, and session frequency, among the nine studies that reported these parameters, the total number of sessions ranged from 1–72 (median: 32), and the total prescribed exercise time ranged from 30 min to 48 h (median: 16 h), per environment. Two trials are notable for their short duration: Calogiuri et al. [27,28] prescribed a total of two 45-min sessions, two days apart, while Miller et al. [17] required participants to complete a total of only one to two 30–50 min sessions within one week, per environment. The remaining seven studies involved a total of 15–72 sessions and 7.5–48 h of prescribed exercise over 3–18 weeks.

##### Exercise Type and Modality

Five trials involved aerobic exercise, specifically running [18,19,29] or walking [17,32]. Two trials combined aerobic exercise (aerobic circuit [30,31]; cycling [27,28]) with resistance training (using elastic bands). Other types of exercise included agility, balance, and strength training [20] and dual-task training [34].

All except two studies matched exercise type between outdoor and indoor conditions. Moslehi et al. [18] compared aerobic exercise outdoors in the form of playing football versus indoors in the form of treadmill running. Abdel-Rahman and Magdy [33] compared “stretch and body weight exercises” outdoors versus “traditional training” indoors, with no further description of exercise protocols. 

Given biomechanical and physiological differences between different exercise modalities (e.g., overground running versus treadmill or cycling versus stationary bike) [35,36,37], exercise type should be considered in the context of exercise modality. Among the eight studies that matched exercise type between outdoor and indoor groups, five also matched the modality [17,19,20,30,31,34], whereas three studies compared outdoor, overground running, cycling, or walking with an indoor, stationary counterpart (i.e., treadmill or stationary bike) [27,28,29,32].

##### Exercise Intensity

Given the positive dose–response relationship between exercise intensity and resulting health benefits [38,39], it is important to identify if and how intensity was prescribed, monitored, controlled, and analyzed (Table 3) in the included studies. In this review, we broadly define two approaches to operationalize exercise intensity: (1) objective parameters (e.g., heart rate reserve); and (2) subjective parameters (e.g., ratings or descriptions of perceived exertion) [39].

Intensity was prescribed with objective parameters in two trials (i.e., percentage of heart rate reserve (% HRR), percentage of maximum heart rate (% HR_max_), percentage of maximal oxygen consumption (VO_2max_)) [18,19], subjective parameters in two trials (i.e., Borg rating of perceived exertion (RPE) scale or descriptions such as a “brisk” pace) [27,28,32], and with both objective and subjective parameters in two trials (%HR_max_, Borg RPE, self-selected intensity) [29,30,31]. Four trials provided no information on any prescribed intensity [17,20,33,34]; one of these studies had participants paced by a peer leader, with no details on the selected pace [17].

For the six included studies that reported exercise intensity prescription, prescribed intensities ranged from light-to-moderate to vigorous, according to evidence-based relative exercise intensity zones for cardiorespiratory exercise [39].

Five studies reported measuring intensity as a control variable. These studies monitored the intensity using objective (i.e., %HRR, %HR_max_, GPS-device-measured distance and speed) [18,19,27,28,29,30] and/or subjective (Borg RPE 6–20) [27,28,29] parameters. These five studies also reported that research staff “controlled”, “regulated”, or “managed” exercise intensity. Two of these studies partially reported results for intensity monitoring, finding no significant differences between outdoor and indoor environments, but unclearly reporting whether the prescribed intensities were achieved [18,27,28]. In contrast, in the trials that did not control for compliance to a prescribed intensity, the effects of exercise intensity may confound those of the exercise environment.

Three studies investigated exercise intensity as an outcome [17,18,30,31]. Moslehi et al. [18] objectively prescribed, monitored, and controlled exercise intensity (using %HRR and GPS-measured distance and speed), and subjectively measured intensity (Borg RPE 1–10) as an outcome. Given that no significant differences between environments were observed for objective exercise intensity, the observed difference in perceived exertion can be attributed to a true effect of the exercise environment, i.e., a decoupling between objective and subjective exercise intensity. Miller et al. [17], whose exercise sessions were paced by a peer leader, did not report prescribing or monitoring the exercise intensity, but objectively measured intensity (actigraphy-measured metabolic equivalents of the task) as an outcome. In this case, it is unclear whether subjective exercise intensity or peer leader pacing—two potential confounders of exercise environment—were adequately controlled in both environments; the observed significant difference in objective exercise intensity may simply indicate poor experimental control of intensity across environments. Finally, two articles corresponding to the same trial reported distinct approaches to prescribing intensity—either a subjective, self-selected intensity [31] or an objective intensity (%HR_max_) [30]. Each of these articles reported the alternative intensity measure as an outcome: Lacharité-Lemieux and Dionne (2016) objectively monitored mean and maximal self-chosen intensity as an outcome [31], whereas Lacharité-Lemieux et al. (2015) measured subjective intensity (Borg RPE 6–20) as an outcome [30]. This suggests either a reporting error or inappropriate inclusion of researcher-controlled variables as outcome variables.

### 3.3. Risk of Bias 

A summary of the risk of bias judgements across the six domains is presented in Figure 2 and discussed below. All but one of the included articles—corresponding to all of the included trials—were judged to have a high risk of bias as their overall assessment.

Three studies [19,33,34] did not make any mention of randomization and were thus assessed to have a high risk of bias arising from the randomization process (Domain 1). Only two of these trials [20,30,31] described the randomization mechanism, and none provided information about allocation concealment. One article [17] was assigned a high risk of bias due to deviations from intended interventions (Domain 2) because it used per-protocol—rather than intention-to-treat—analysis, with an unclear impact on the result; no articles explicitly described analyses as intention-to-treat, complete case, or per-protocol. Five trials were judged to have a high risk of bias due to missing data (Domain 3) because the authors provided no information on the extent of missing data, numbers of participants analyzed for results [18,20,33,34], or reasons for missing data [17]. Seven studies were judged to have “some concerns” for risk of bias in the measurement of the outcome (Domain 4) because either the outcome assessor and/or the participant was aware of the prescribed intervention (outdoor or indoor exercise), and subjective, judgement-based outcome measures were included [17,27,28,29,32,33,34]. Four articles [27,30,31,32] showed a high risk of bias in the selection of the reported results (Domain 5) from statistical analyses [32] or from multiple eligible outcome measures within the outcome domain [27,30]. The single crossover trial [17] was assessed to have a high risk of bias due to inappropriate timing of measurement sessions to account for carryover effects (Domain S). Unclear reporting frequently contributed to “some concerns” in the risk of bias judgements.

Other risks of bias or confounding included analyses not accounting for baseline outcome values [34], small sample size [17,18,19,20,27,28,30,31,33], short intervention duration (≤two sessions per environment) [17,27,28], differences in exercise protocols beyond the physical environment (e.g., exercise type or modality, social interaction) [18,19,27,28,33], lack of control of exercise volume or intensity [17,32], lack of controlling for multiple comparisons (adjustments performed in only three studies [18,19,29]), and absence of prospective power calculation to determine sample size (only reported in three articles [17,18,27]; with one other trial basing sample size on a prior study [30,31]).

### 3.4. Study Outcomes

Outcomes and corresponding measurement tools are listed in Appendix A. Six trials assessed psychological health outcomes [17,27,28,29,30,32,34], eight trials assessed physical health outcomes [18,19,20,28,29,31,33,34], and three trials assessed physical activity behaviour [17,27,30].

Given the extensive outcome heterogeneity, we classified discrete outcomes (e.g., positive engagement, positive affect, exercise enjoyment) into broader outcome categories (e.g., Positive Emotions) within each outcome group (e.g., Psychological) (Table A1). Most of the discrete outcomes were assessed in only one or two trials. The outcome categories assessed in the most trials included weight (four trials) and body composition (four trials). Although four trials measured various hormones and neuropeptides, no discrete outcome in this category was assessed in more than one trial. Similar affective outcomes were assessed in three trials, but with different scales that were sometimes incompatible, including dimensional (e.g., Feeling Scale, Felt Arousal Scale) and categorical (e.g., Physical Activity Affective Scale, Exercise-Induced Feeling Inventory) measures.

Despite including the most outcomes and comparisons, only Lacharité-Lemieux et al. [30,31] explicitly declared primary and secondary outcomes. Although other trials reported multiple main outcomes, they did not clearly specify the relative priority among these outcomes.

### 3.5. Results of Included Studies

The results of individual studies are presented in Appendix A. Meta-analysis was not performed due to the minimal overlap of outcomes across trials. Instead, in Table 4, we synthesized the results qualitatively by grouping discrete outcomes into broader categories and indicating the direction of significant group differences in each outcome category. All findings of statistically significant differences between outdoor and indoor conditions favoured outdoor exercise.

The low number of studies, heterogeneity of outcomes, and heterogeneity of participant populations hindered the interpretation of any strong trends in the results. Given this limitation, we used vote counting to summarize the direction of significant effects in each outcome group.

#### 3.5.1. Psychological Health Outcomes

Psychological health outcomes were measured in six trials, including a total of 35 comparisons between outdoor and indoor conditions. This group comprised 10 outcome categories, with 6 of these assessed in 2 or more trials. Significant between-group differences in treatment effects—all favouring the outdoor condition—were found for 7 of the comparisons (25%), including 4 of the outcome categories (positive emotions, tranquillity, restoration, and motivation).

#### 3.5.2. Physical Health Outcomes

Physical health outcomes were measured in eight trials, including 56 comparisons between outdoor and indoor conditions. This group included anthropometric, physiological, and physical fitness outcomes, divided into 13 outcome categories. Significant differences—all favouring the outdoor condition—were observed for 15 of the comparisons (27%) over 8 of the outcome categories, including 3/16 (19%) comparisons for anthropometric outcomes, 5/16 (24%) comparisons for physiological outcomes, and 7/19 (37%) comparisons for fitness outcomes.

#### 3.5.3. Physical Activity Behaviour

Physical activity behaviour was measured in three trials, across three outcome categories, for a total of eight comparisons between outdoor and indoor conditions. Significant, outdoor-favourable effects were found for three comparisons (38%) over the three outcomes categories. Notably, to measure physical activity level, actigraphy was used in one trial (one comparison), whereas self-report questionnaires were employed for the remaining four comparisons (across two trials), including the single significant finding.

#### 3.5.4. Overall Directionality of Findings

In total, 99 comparisons were made between outdoor and indoor environments across all 10 trials, including 74 non-significant comparisons (74%), 25 comparisons favouring outdoor exercise (25%), and none favouring indoor exercise.

Among the outcome categories observed in multiple trials (15 categories among 27 total), only one category—mobility—showed significant effects for the majority of comparisons (5/7) and in more than one trial (n = 3). Most of these significant comparisons (4/6) were derived from two trials in older adults [20,34]. 

## 4. Discussion

### 4.1. Summary of Findings

The objective of this review was to compare the effects of longitudinal exercise in an outdoor environment, as compared to an indoor environment, on psychological and physical health, as well as physical activity behaviour. The collective evidence from the ten included longitudinal trials was minimal and uncertain regarding the potential benefits of an outdoor versus indoor exercise environment. 

The included trials varied widely in participant populations, exercise protocols, outcomes, and clarity of reporting. These factors, together with the low number of studies and high risk of bias in all studies, hindered the synthesis, interpretation, and generalization of findings. We acknowledge that conducting longitudinal exercise trials is difficult and resource-intensive, which likely contributes to the low number of longitudinal studies.

While a minority of the comparisons suggested benefits from outdoor exercise among some health and behaviour outcomes, there was insufficient overlap in outcomes to make any strong conclusions. It is noteworthy that all findings of statistically significant differences between outdoor and indoor settings favoured outdoor exercise. Further evidence is required to clarify whether these results accurately represent treatment effects, or result from bias arising within (e.g., unblinded outcome assessment, selective reporting, multiple testing) and/or across studies (e.g., publication bias).

### 4.2. Quality of Evidence

Our search revealed that few longitudinal studies have been conducted to investigate the potential benefits of an outdoor exercise environment. Furthermore, all the included studies—and almost all of the included articles—were judged to have an overall high risk of bias. Therefore, study results should be interpreted with a degree of caution.

Risk of bias arose from various sources: lack of information on the randomization method, allocation concealment, and groupwise baseline differences (Domain 1); no blinding of participants and intervention facilitators (impossible in this context), together with a failure to report deviations from intended interventions and use of intention-to-treat, complete case, or per-protocol analyses (Domain 2); failure to report numbers of participants analyzed and extent of missing data (Domain 3); no blinding of outcome assessors, together with subjective, participant-reported outcomes (Domain 4); lack of pre-registration of analysis plans to help prevent selective reporting; incomplete reporting of data for all outcomes outlined in methods; and incomplete reporting of multiple outcome measures within an outcome domain and/or multiple analyses of the data (Domain 5).

Other notable risks of bias or confounding included small samples, lack of prospective power calculation to determine sample size, failure to control for multiple comparisons, differences in experimental conditions between environments (e.g., exercise type, social interaction), and analyses not accounting for baseline values, altogether decreasing confidence in reported effects. We acknowledge that two of the included studies were described as pilot studies [17,27,28], and so may have refrained from adjusting for multiple testing to reduce the risk of type II error at the expense of increasing the risk of type I error. 

Additionally, unclear reporting hindered data extraction, risk of bias assessments, and interpretation of results. All but one trial [30,31] failed to explicitly declare the priority of outcomes. Six studies lacked detail in reporting methods, such as descriptions of exercise environment [29,32,33,34], exercise protocol [17,33], outcome measures [33], and statistical analyses (unclear type of ANOVA employed [18,29,33]). Four trials also failed to report groupwise baseline participant data [27,28,32,34]. Reporting of outcome data and statistical test results was often incomplete or unclear; for instance, articles omitted groupwise numerical data and/or statistical significance tests for some outcomes, or, within the ANOVA model, confused or did not clearly distinguish marginal effects and group-by-time interactions [29,30,31,32,34]

Finally, although these studies investigated the effects of longitudinal exercise interventions, only one study assessed the duration of effects by performing follow-up measures. However, for this study, the initial intervention only involved two exercise bouts over a three-day period [27,28].

In the context of outdoor versus indoor exercise trials, we acknowledge that some risks of bias can be difficult or impossible to avoid, such as participant blinding or between-environment protocol differences. Nevertheless, most of the highlighted risks of bias are avoidable through rigorous study design and reporting, as outlined in Section 4.5.

### 4.3. Limitations and Biases in the Current Review

First, our eligibility criteria may have excluded some studies relevant to the health effects of an outdoor or natural exercise environment. For practical reasons, our search was limited to English language articles, thus excluding any relevant studies unavailable in English. Furthermore, the current review focused exclusively on longitudinal trials; we are preparing a separate review on acute trials. Additionally, the scope of our review was limited to comparisons of outdoor environments versus indoor, non-natural environments, thus excluding numerous studies comparing various qualities of outdoor environments (natural, built, mixed) without an indoor control, or studies comparing outdoor, natural environments (outdoor green exercise) versus indoor environments with actual or simulated nature elements (indoor or virtual green exercise). Such studies would need to be considered together with the studies included herein to more comprehensively assess the experimental evidence for the benefits of exercising in an outdoor and/or natural versus indoor and/or built exercise environment. Secondly, we directly extracted results regarding the significance and direction of comparisons between outdoor and indoor exercise as reported in each article; we did not perform any independent quantitative analysis on outcome data to verify the magnitude or statistical significance of these comparisons. The use of vote counting based on the direction and significance of effects also has several limitations, including potential misrepresentation of effects from underpowered studies, not accounting for the magnitude of effects, and not accounting for differences in study sample sizes [40]. 

### 4.4. Comparison to Previous Reviews

Lahart et al. [16] systematically reviewed both longitudinal and acute studies comparing exercise in outdoor natural environments or in the presence of simulated nature versus indoor settings without nature exposure. Our review criteria differed from Lahart et al. [16] in that (1) we focused exclusively on longitudinal trials and (2) we defined the intervention condition as exercise in any outdoor environment, excluding exercise with virtual exposure to nature (e.g., images, video, or virtual reality). The current review included all three longitudinal trials (corresponding to five articles) identified by Lahart et al. [16], as well as seven additional trials, six of which were published since 2019. As a result, the present review included more comparisons of many of the previously identified outcomes, and multiple previously uninvestigated psychological and physical outcomes, such as self-efficacy, motivation, hormones and neuropeptides, mobility, and balance. Lahart et al. [16] performed meta-analyses of 10 psychological and physical outcomes (each assessed in two longitudinal trials) and qualitative synthesis for the remaining outcomes (using vote counting based on the direction of significant effects). In contrast, we exclusively performed qualitative synthesis due to minimal inter-study overlap in outcomes. Additionally, Lahart et al. [16] included RPE and objective exercise intensity (i.e., HR-based) as outcomes in their meta-analyses; while doing so, they conflated exercise intensity as a control variable (i.e., confirmation of adherence to protocol) with exercise intensity as an outcome. In our review, we restricted our synthesis to health-related outcomes, thus excluding RPE and exercise intensity; we reviewed exercise intensity only in the context of exercise protocols, and we distinguished between intensity as a control variable and intensity as an outcome.

Despite the above-noted differences, our overall findings and conclusions for longitudinal trials remain similar to those of Lahart et al. [16]: inconclusive evidence for the benefits of longitudinal outdoor exercise, high risk of bias in most articles, and a need for more rigorous trials and more thorough reporting.

### 4.5. Recommendations for Future Research

The lack of high-quality research on the potential benefits of longitudinal outdoor exercise highlights the need for more rigorously designed long-term trials comparing the health and behaviour effects of exercise outdoors versus indoors. Specifically, future randomized trials should incorporate pre-registration of study protocols and statistical analysis plans; prospective analysis of power and sample size; adequate randomization procedures; allocation concealment; blinding of outcome assessors (where feasible); standard, validated, and reliable outcome measures to facilitate between-study comparison; statistical adjustment for multiple comparisons (where appropriate); and outcome measurement not only at the trial end to evaluate immediate treatment effects, but also after a follow-up period to evaluate the duration of effects. Equally, future studies should aim to clearly report methods and results following standard reporting guidelines (e.g., Consolidated Standards of Reporting Trials {CONSORT} [41]). 

When implementing and reporting the exercise protocol, researchers should specify: (1) how and what intensity is prescribed (using either objective or subjective parameters), (2) how intensity is monitored (using either objective or subjective measures), (3) any steps taken to promote adherence to the prescribed intensity across both conditions (i.e., to “control” the intensity), and (4) whether intensity is analyzed as a control variable or as an outcome variable, avoiding the inappropriate analysis of researcher-controlled variables as outcome variables.

The reviewed longitudinal trials overlooked or minimally investigated several notable outcomes theorized [42,43,44] and/or demonstrated—in acute trials [45,46,47]—to be benefited by nature exposure. Future longitudinal trials should investigate such outcomes, including objectively measured cognitive domains (e.g., working memory), physiological stress markers (i.e., blood pressure and cortisol), immune function (e.g., natural killer T cell activity), autonomic nervous activity (e.g., heart rate variability), brain activity, and physical activity level (especially with objective measures).

Preliminary research from acute trials and observational studies suggests that exercise responses may vary as a function of the type of outdoor and indoor environment (e.g., forest trail, urban walkway, well-lit gym), the particular population subgroup, the type of exercise, and the interaction between these factors [48,49,50,51]. Future research should explore the effects of various types of exercise and outdoor environments in various population subgroups, including clinical populations (e.g., mental health conditions, chronic stress, obesity) and healthy individuals from different demographics.

## 5. Conclusions

This systematic review identified 10 longitudinal trials (corresponding to 12 articles) investigating psychological health, physical health, and physical activity behaviour responses to outdoor versus indoor exercise. Overall, there was limited evidence for significant additional effects of an outdoor exercise environment. Included trials assessed a diversity of outcomes with insufficient overlap to perform meta-analysis. Across the 10 trials, 25 of the 99 comparisons between outdoor and indoor exercise environments found benefits from outdoor exercise, while the remainder found no significant differences. Extraction, synthesis, and interpretation of findings were impeded by the heterogeneity of outcomes, participant populations, and methods, unclear reporting, and an overall high risk of bias. Future more robustly designed longitudinal trials are needed to clarify the potential benefits of chronic exercise outdoors versus indoors. 

## Figures and Tables

**Figure 1 ijerph-20-01669-f001:**
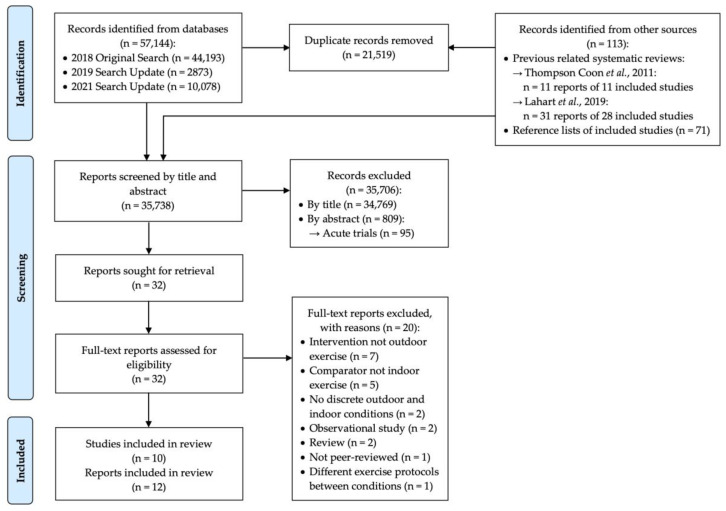
PRISMA flow diagram of article search and screening process.

**Figure 2 ijerph-20-01669-f002:**
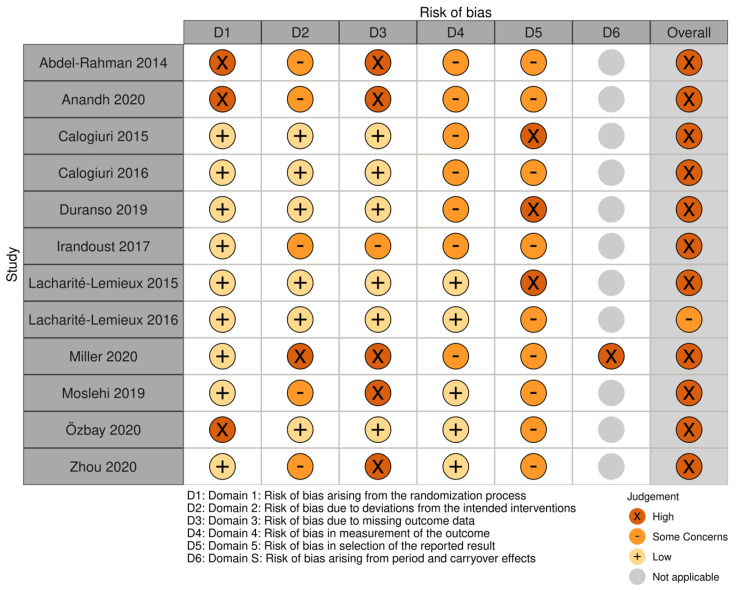
Summary of domain-level risk of bias judgements for each included article.

**Table 1 ijerph-20-01669-t001:** Master search strategy. Shown is the strategy designed in Ovid MEDLINE and adapted for other databases. *: indicates truncation to capture multiple suffices of the base search term. Abbreviations: ti: title; ab: abstract.

	Search terms	Category
1	(green exercis* or green gym* or blue exercis* or blue gym* or ecotherap*).ti,ab.	Outdoor Exercise
2	(exercis* or physical activit* or walk* or physical fit* or run* or athlet*).ti,ab.	Exercise
3	(outdoor* or outside* or park* or greenspace* or green space* or bluespace* or blue space* or natural environment* or nature or forest* or biodivers* or horticultur*).ti,ab	Outdoors
4	(indoor* or inside* or laboratory or gym* or home* or buil*).ti,ab.	Indoors
5	2 and 3	
6	1 or 5	
7	4 and 6	
8	limit 7 to English language	

**Table 2 ijerph-20-01669-t002:** PICOS eligibility criteria for the current systematic review.

PICOS	Inclusion Criteria
Population	No restrictions
Intervention	Two or more exercise bouts in an outdoor environment
Comparison	Two or more exercise bouts performed indoors with no exposure to actual or simulated nature
Outcome	Primary: at least one outcome related to physical or psychological health Secondary: any outcomes related to physical activity behaviour
Study design	Randomized or non-randomized trials

**Table 3 ijerph-20-01669-t003:** Operational definitions of terms related to regulating and measuring exercise intensity.

Term Related to Exercise Intensity	Operational Definition
Prescribe	Were participants instructed to exercise at a target intensity (either objective or subjective)?
Monitor	Did instructors measure exercise intensity (using either objective or subjective measures)?
Control or Regulate	Did instructors advise participants to maintain or adjust their intensity as necessary to match the target?
Measure/analyze as an outcome or as a control variable	Was intensity treated as an outcome (a dependent variable upon which the effect of the intervention is being investigated), as a control variable (intended to be equal to the prospectively defined target and equal in both groups), or inappropriately as both?

**Table 4 ijerph-20-01669-t004:** Summarized findings of included studies.

Outcome Category			Author, Year											
			Abdel-Rahman et al., 2014 [33]	Anandh, Varadha-rajulu, & Alate, 2020 [34]	Calogiuri, Nordtug, & Weydahl, 2015 [Intervention] [27]	Calogiuri et al., 2016 [28]	Duranso, 2018 [32]	Irandoust & Taheri, 2017 [29]	Lacharité-Lemieux, Brunelle, & Dionne, 2015 [30]	Lacharité-Lemieux & Dionne, 2016 [31]	Miller et al., 2020 [17]	Moslehi, Moslehi, & Khalvati, 2019 [18]	Özbay et al., 2020 [19]	Zhou et al., 2020 [20]
PSYCHOLOGICAL		Affective Valence							NS 2/2					
		Positive Emotions			OE 1/2	OE 2/3			NS					
					NS 1/2	NS 1/3								
		Depression						NS	NS					
		Affective Activation							NS 2/2					
		Tranquility				NS 2/2			OE					
		Restoration			OE 2/2	OE 2/2								
		Energy							NS					
		Fatigue							NS		NS 2/2			
		Self-Efficacy and Self-Determination		NS			NS 2/2				NS 3/3			
		Motivation					NS 2/2				OE 1/6			
											NS 5/6			
PHYSICAL	Anthro-pometric	Weight						NS		NS		OE	NS	
		Body Composition						NS 3/3		NS 6/6		OE 2/2	NS	
	Physio-logical	Systolic Blood Pressure				NS				NS				
		Diastolic Blood Pressure				OE				NS				
		Plasma Lipids								NS 4/4			OE 1/3	
													NS 2/3	
		Glucose and Insulin Profile								NS 3/3				
		Hormones and Neuropeptides				OE 1/3		OE				OE	NS 2/2	
						NS 2/3								
	Physical Fitness	Flexibility	OE											
		Mobility	OE	OE 1/2										OE 3/4
				NS 1/2										NS 1/4
		VO_2max_								NS		NS		
		Muscle Strength								NS 3/3				
		Muscle Endurance								OE 1/3				
										NS 2/3				
		Balance												NS 3/3
BEHAVIOUR		Future Exercise Intention			OE									
		Physical Activity Level			OE 1/4				NS		NS			
					NS 3/4									
		Exercise Adherence							OE					

**Outcome Categories:** Refer to Table A1 for lists of discrete outcomes included within each outcome category. **Symbols and Acronyms:** ‘OE’: A statistically significant group difference was reported for one or more outcomes included in this category; the direction of the effect favoured outdoor exercise. ‘NS’: One or more outcomes included in this category were measured in the study; no statistically significant group differences were reported. [blank]: No outcomes in this category were measured in the study. Fractions represent the number of comparisons with the indicated result, out of the total number of comparisons measured in this category.

## Data Availability

The following data are available upon request by contacting the corresponding author: a detailed review protocol; a search tracking spreadsheet; data extraction forms; and risk of bias decisions and justifications.

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
