# Peer review of "The Effects of Outdoor versus Indoor Exercise on Psychological Health, Physical Health, and Physical Activity Behaviour: A Systematic Review of Longitudinal Trials"

_ijerph, 2023, doi:10.3390/ijerph20031669_

Round 1
Reviewer 1 Report
The paper entitled, “The Effects of Outdoor versus Indoor Exercise on Psychological 2 Health, Physical Health, and Physical Activity Behaviour: A Systematic Review of Longitudinal Trials” is an elaborative and backbreaking work of research synthesis by the authors. But the execution of this research starting from motive to analysis seems to be derailed in between. I recommend some important inputs which would make this paper better to understand and easy to cite.
1. Introduction is unnecessarily too lengthy, it should be reduced and I don’t fine the importance of Figure 1 in relation to the hypothesis and outcome of this paper.
2. There should be clear flowchart regarding search criteria and selection of the studies, although authors have drawn PRISMA diagram. I mean make it comprehensible by drawing arrows and extraction points.
3. Authors have compared and analysed effects of exercise in outdoor environments versus indoor environments on psychological health, physical health, and physical activity behaviour. But in results section, the inferences are incomplete, inconclusive and inconceivable. This section should clearly define the outcome.
4. I don’t find a need to write so many recommendations out of this. One para not more than 8-10 lines is sufficient.
5. Authors have mentioned themselves in conclusion, …”The majority of comparisons between outdoor and indoor exercise environments found no significant differences, while a minority of comparisons found benefits from outdoor exercise. Extraction, synthesis, and interpretation of findings were impeded by heterogeneity of outcomes, participant populations, and methods, unclear reporting, and an overall high risk of bias….
In this scenario its citation and readability will be hugely reduced….so I suggest the authors to write conclusion in different style in more impressive way.
Author Response
Point 1: Introduction is unnecessarily too lengthy, it should be reduced and I don’t fine the importance of Figure 1 in relation to the hypothesis and outcome of this paper.
Response 1: Thank you for this feedback; we have revised accordingly. We have removed text about observational studies on greenspace and physical activity participation in paragraph 2. We deemed the other information important to provide background and build rationale, progressing from physical activity and its health benefits (paragraph 1), the broader body of research on health effects of nature exposure in general and the potential benefit of nature exposure combined with exercise (paragraph 2), the previous, related systematic reviews (paragraphs 3 and 4), and the motivation for updating these reviews (paragraph 5).
We have kept Figure 1 to demonstrate the increase in green exercise research over the past decade, as part of the rationale for conducting systematic reviews related to green exercise. If recommended by the reviewer, we could move this figure to the Appendix.
Point 2: There should be clear flowchart regarding search criteria and selection of the studies, although authors have drawn PRISMA diagram. I mean make it comprehensible by drawing arrows and extraction points.
Response 2: We appreciate this suggestion. We are confused by the reviewer’s recommendation of “drawing arrows and extraction points”. We ask for clarification on specifically what needs to be added to the PRISMA diagram and/or to other sections of the manuscript. We believe that we have adequately followed PRISMA guidelines for presenting search strategy (item 7) and selection process (items 8 and 16), as indexed in our PRISMA checklist (Supplementary Table S1) and explained below.
For “search criteria”, while we do not have a flow chart, we have included: (i) the search strategy in Section 2.2 and Table 1, along with the detailed search strategy for each database in Supplementary Table S1; and (ii) eligibility criteria in the main text (Section 2.3) and in Table 2.
For “selection of the studies”, we have presented the PRISMA flow diagram along with a text summary in Section 3.1.
Point 3: Authors have compared and analysed effects of exercise in outdoor environments versus indoor environments on psychological health, physical health, and physical activity behaviour. But in results section, the inferences are incomplete, inconclusive and inconceivable. This section should clearly define the outcome.
Response 3: Thank you for this feedback. We feel that we have done our best to summarize and consolidate the minimal and heterogeneous evidence available. We seek clarification from the reviewer as to how we can increase the clarity of the Results and/or Discussion sections given the state of the evidence.
In the Results section, specifically Section 3.5, we have tried to summarize the results of included studies within each of the three main outcome groups (psychological health, physical health, and physical activity behaviour). Given minimal outcome overlap between studies, we used outcome groupings and vote-counting to help summarize results and highlight any trends. We have tried to refrain from any inferences until the Discussion.
In case the reviewer’s above point is referring to the Discussion, we note the following. In the Discussion, Section 4.1 (Summary of Findings), we indeed concluded that the collective evidence for benefits of outdoor versus indoor exercise is (1) minimal and (2) uncertain or “inconclusive”. We explained how this conclusion arises from the limitations of the existing body of evidence: the paucity of studies, the heterogeneity of study characteristics and outcomes, and the high risk of bias.
With regards to “clearly defin[ing the outcome]”, the three main groups of outcomes of this review—psychological health, physical health, and physical activity behaviour—are described in the first paragraph of Results Section 3.4 (Study Outcomes); we have re-worded this sentence in hopes of making it clearer. These three outcome groups are also outlined in Methods Section 2.3. Additionally, as described in Methods Section 2.6 and Results Sections 3.4 and 3.5 and as shown in Table 4, we have grouped discrete outcomes (e.g., Positive Affect) into broader outcome categories (e.g., Positive Emotions), within each major outcome group (e.g., Psychological Health). These outcome categories are defined in Appendix Table A1.
Point 4: I don’t find a need to write so many recommendations out of this. One para not more than 8-10 lines is sufficient.
Response 4: Thank you. In response to this suggestion, we have condensed this section significantly. However, we largely retained the first two paragraphs as we feel that they outline important measures to mitigate risk of bias.
Point 5: Authors have mentioned themselves in conclusion, …”The majority of comparisons between outdoor and indoor exercise environments found no significant differences, while a minority of comparisons found benefits from outdoor exercise. Extraction, synthesis, and interpretation of findings were impeded by heterogeneity of outcomes, participant populations, and methods, unclear reporting, and an overall high risk of bias….
In this scenario its citation and readability will be hugely reduced….so I suggest the authors to write conclusion in different style in more impressive way.
Response 5: We acknowledge that the first sentence may have been confusing. To clarify, when we state, “The majority of comparisons…”, we are referring to comparisons performed within each included study; we are not referring to ourselves. We have re-worded this sentence to reduce any confusion and any downplaying of the results.
Reviewer 2 Report
Overall, in the discussion section, the authors need to acknowledge the ramifications of systematic reviews and meta-analyses.
In particular: a. Any review study “acts” as a gateway of what is and what is not “valid”. Therefore, I propose that the authors comment that the selection of studies based on the focus of the study, inevitably, highlights some studies while “ignoring” others that may have merit. Such comments may be inserted in lines 449-452
b. In commenting about the low number of studies and risk bias (lines 454 – 457) I would suggest that the authors consider that it is easier to judge and critique research than doing research. In particular, any type of longitudinal trial is difficult to be realized and it will certainly involve some type of bias. This is especially true for real –life interventions, such as exercising outdoors, in contrast to laboratory experiments. I suggest that the authors acknowledge these issues adding respective materials in line 454 – 457 and in the quality of evidence section.
c. In the quality of evidence section, the authors point out several “deficiencies” of the studies reviewed such as “lack of information of the randomization method”, small samples groupwise baseline differences and differences in experimental conditions. In my view, these criteria stem and are more akin to medical laboratory based research and their endorsement lead to unnecessary “medicalization” of research. Further, this ignores basic real-life conditions. For example, regarding randomization, in a study having participants who exercise in natural vs built environments, I wonder, randomize on what? Some participants may have extra weight, some may not. Some may have a recent exercise history some may not. Some may have a childhood sport history and some may not. Some may have peers who exercise and some may not. Some may hold more favorable attitudes to exercise and some may not. Some may have been more acquainted with exercising indoors and others may be more acquainted with exercising outdoors. The list can be endless. I do not think that adopting uncritically such criteria for judging research aids to the advancement of the field. I suggest that the authors should be more mediocre in their “judgement” of studies and acknowledge that these criteria not be that relevant to physical activity interventions and at the same time destruct from the applicability of this research.
d. I fully disagree with the authors’ suggestion to prescribe, monitor and match exercise intensity (lines 561 – 567). This suggestion is out of context because exercising outdoors is simply different than exercising indoors. Depending on the mode, the intensity of exercising outdoors can be intermitted – e.g. playing football with friends at the park, while exercising indoors can be of stable intensity – e.g. running on a ergometer. In suggesting that research should match the intensity in different environments we lose sight and I suggest that the authors eliminate this paragraph.
Minor comments
Line 38. Please replace “myriad” with “multiple”
Author Response
Point A: Any review study “acts” as a gateway of what is and what is not “valid”. Therefore, I propose that the authors comment that the selection of studies based on the focus of the study, inevitably, highlights some studies while “ignoring” others that may have merit. Such comments may be inserted in lines 449-452
Response A: Thank you for highlighting this limitation. We have acknowledged this limitation in Section 4.3 (Limitations and biases in the current review). We have elaborated on our original point to more fully address this limitation.
Point B: In commenting about the low number of studies and risk bias (lines 454 – 457) I would suggest that the authors consider that it is easier to judge and critique research than doing research. In particular, any type of longitudinal trial is difficult to be realized and it will certainly involve some type of bias. This is especially true for real –life interventions, such as exercising outdoors, in contrast to laboratory experiments. I suggest that the authors acknowledge these issues adding respective materials in line 454 – 457 and in the quality of evidence section.
Response B: We concur with the reviewer that conducting longitudinal interventions is difficult and likely limits the number of studies performed. We added a note to acknowledge this issue in Section 4.1 in paragraph 2. We also concur with the reviewer that avoiding bias can sometimes be challenging in human interventions outside of ideal laboratory conditions. We have added a note to acknowledge this reality at the end of Section 4.2 (Quality of Evidence), which we felt was the most appropriate location. However, we also assert that most of the highlighted risks of bias can be avoided in the study design and reporting phases.
Point C: In the quality of evidence section, the authors point out several “deficiencies” of the studies reviewed such as “lack of information of the randomization method”, small samples groupwise baseline differences and differences in experimental conditions. In my view, these criteria stem and are more akin to medical laboratory based research and their endorsement lead to unnecessary “medicalization” of research. Further, this ignores basic real-life conditions. For example, regarding randomization, in a study having participants who exercise in natural vs built environments, I wonder, randomize on what? Some participants may have extra weight, some may not. Some may have a recent exercise history some may not. Some may have a childhood sport history and some may not. Some may have peers who exercise and some may not. Some may hold more favorable attitudes to exercise and some may not. Some may have been more acquainted with exercising indoors and others may be more acquainted with exercising outdoors. The list can be endless. I do not think that adopting uncritically such criteria for judging research aids to the advancement of the field. I suggest that the authors should be more mediocre in their “judgement” of studies and acknowledge that these criteria not be that relevant to physical activity interventions and at the same time destruct from the applicability of this research.
Response C: Thank you for these insights which we cannot address in this context. Our pre-defined objective for this review was to compare the effects of controlled trials of exercise in outdoor versus indoor environments. In the field of exercise trials, the aforementioned study design criteria (e.g., randomization, a prospective power calculation to determine sample size, and equalizing experimental conditions apart from the independent variable), as well as clear reporting of methods and results, are discussed extensively in the literature as relevant and critical factors to mitigate biases and confounds, and to maintain the transparency, validity, and strength of the evidence.
Point D: I fully disagree with the authors’ suggestion to prescribe, monitor and match exercise intensity (lines 561 – 567). This suggestion is out of context because exercising outdoors is simply different than exercising indoors. Depending on the mode, the intensity of exercising outdoors can be intermitted – e.g. playing football with friends at the park, while exercising indoors can be of stable intensity – e.g. running on a ergometer. In suggesting that research should match the intensity in different environments we lose sight and I suggest that the authors eliminate this paragraph.
Response D: Thank you for the insight. We believe that our suggestion aligns with the context of the current review, as specified in the pre-registered review protocol. The pre-defined objective of the review was to compare the effects of controlled trials of exercise in outdoor versus indoor environments. As specified in our eligibility criteria, our independent variable of interest was the exercise environment, with the outdoor environment as the intervention, the indoor environment as the control treatment, and all other conditions (e.g., exercise type, duration, intensity; social setting) matched as much as possible. Such a study design attempts to isolate the effects of the independent variable (the physical exercise environment) from potentially confounding variables (e.g., exercise intensity). If comparing outdoor versus indoor exercise with additional, non-environment differences in exercise conditions, it is impossible to identify which variables contribute to any observed effects on the outcomes.
We acknowledge that it may not always be possible or feasible to perfectly match outdoor and indoor exercise conditions, and emphasize the idea of matching conditions as much as possible. In the example given by the reviewer, the ideal comparator for outdoor football with friends would be indoor football with friends, while a reasonable comparator, assuming an indoor turf or gym is unavailable, might be indoor treadmill running with intermittent changes in intensity, alongside friends. Indeed, among the studies in the review, Moslehi et al. [1] compared outdoor football versus indoor treadmill running; while the researchers attempted to control and equalize intensity across both groups, the effects of the environment are still confounded by the exercise type (football versus running) and social context.
Minor comments:
Line 38. Please replace “myriad” with “multiple”
Response: Thank you for this suggestion. We made this change.
Reviewer 3 Report
Thank you for this very important work. The procedure you used is very well explained and the results are well documented. I enjoyed reading the manuscript.
I have no suggestions for any improvement of the manuscript because from my viewpoint, everything is well-documented and the information you give is needed in order to understand the conclusions.
My main concern with the manuscript is still not its quality but that it may not present enough new studies since the review of Lahart et al., 2019, also published in the same journal. Also, the result of the review does not lead to clear answers about the importance of the environment when doing exercise. But this is not the fault of the authors. On the other hand, the authors make quite a few suggestions how to improve future research initiatives in that field.
Author Response
Thank you for your feedback. We understand the reviewer’s concern regarding the limited number of new studies (n = 7) since the review of Lahart et al., 2019 and the inconclusive evidence in the current review. We concur that both are beyond our control, and simply represent the state of the current body of literature.
Round 2
Reviewer 1 Report
Authors of the paper have shown that they are not convinced with my review and sidetracked each and every pointer that I have raised. In that scenario, I suggest the authors to reconsider these once again.
Point 1: Introduction is unnecessarily too lengthy, it should be reduced.
It is still of the same length.
Yes, figure 1 should be moved to supplementary files.
Point 2: There should be clear flowchart regarding search criteria and selection of the studies, although authors have drawn PRISMA diagram. I mean make it comprehensible by drawing arrows and extraction points.
For more clarification, I have appended a flow diagram with arrows and extraction points.
Point 3: Result section should neatly define the outcome.
Point 4:I don’t find a need to write so many recommendations out of this. One para not more than 8-10 lines is sufficient.
It has not been done; I really find it overtly nugatory.
Point 5: Authors have mentioned themselves in conclusion, …”The majority of comparisons between outdoor and indoor exercise environments found no significant differences, while a minority of comparisons found benefits from outdoor exercise. Extraction, synthesis, and interpretation of findings were impeded by heterogeneity of outcomes, participant populations, and methods, unclear reporting, and an overall high risk of bias….
In this scenario its citation and readability will be hugely reduced….so I suggest the authors to write conclusion in different style in more impressive way.
Same lines have been added in the conclusion. Remove these controversial lines.

Author Response
Reviewer Follow-Up Overall Comment: Authors of the paper have shown that they are not convinced with my review and sidetracked each and every pointer that I have raised. In that scenario, I suggest the authors to reconsider these once again.
Author Response: We’d like to emphasize that we did our best to address each of the reviewer’s suggestions while maintaining elements that we feel are important for the background, methods, and findings of the review. We’d also like to clarify that the first revised version of the manuscript was uploaded as a Microsoft Word document only, so as to display tracked changes and comments. Based on the reviewer’s second round of comments, especially Point 2, we believe that the reviewer may have looked at the PDF version of the manuscript, which would not have shown any of the revisions made.
We have uploaded the second revised version of the manuscript in both Microsoft Word and PDF formats; however, we advise the reviewer to look at the Microsoft Word version so that they can see the tracked changes.
Reviewer Follow-up 1: It is still of the same length.
Yes, figure 1 should be moved to supplementary files.
Author Follow-up 1: We reiterate that, in the first round of revisions, we removed one sentence in paragraph 2, and justified the importance of the remaining content. We have now removed an additional sentence from paragraph 2. If there are other specific elements that the reviewer feels are unnecessary, please indicate them. For comparison, we note that the length of our introduction (593 words) is shorter than that of both previous reviews on outdoor versus indoor exercise: 757 words for that of Thompson-Coon et al. (2011); 939 words for that of Lahart et al. (2019).
We have moved Figure 1 to the Appendix, and changed its title to Figure A1. We also adjusted the numbering in the other figure titles accordingly. We chose to move the figure to the Appendix, rather than Supplementary Files, so that, if desired, readers may quickly access it within the same document once they see it mentioned in the Introduction.
Reviewer Follow-Up 2: For more clarification, I have appended a flow diagram with arrows and extraction points.
Author Follow-Up 2: We apologize as we believe that there was an error in the PDF version of the manuscript. We noticed that the PRISMA flow diagram (Figure 2 in the original and first revised manuscripts) did not display properly in the PDF version of the manuscript; specifically, the flow diagram lacked arrows and boxes. We believe that this might explain the issues raised by the reviewer in Point 2. Despite this, the PRISMA flow diagram appeared fine in the Microsoft Word files and in the separate Figures file; we believe that there had been an error when converting the manuscript from Microsoft Word to PDF formats.
We have corrected the flow diagram such that it now appears properly both in the Microsoft Word and PDF versions of the second revised manuscript.
Reviewer Follow-Up 3: Result section should neatly define the outcome.
Author Follow-Up 3: We again highlight that the three main groups of outcomes of this review—psychological health, physical health, and physical activity behaviour—are described in the first paragraph of Results Section 3.4 (Study Outcomes). Results within each of these outcome groups are summarized separately in Results Sections 3.5.1–3.5.3.
Please specify any further actions required to make the results neater.
Reviewer Follow-Up 4: It has not been done; I really find it overtly nugatory.
Author Follow-Up 4: We reiterate that we significantly condensed this section (from 604 words to 363 words) in the first round of revisions. We have retained the recommendations that we feel are important and useful for future studies to address the limitations and gaps of the reviewed studies.
Reviewer Follow-Up 5: Same lines have been added in the conclusion. Remove these controversial lines.
Author Follow-Up 5: We reiterate that, in response to Point 5, we previously re-worded the first sentence to increase clarity during the first round of revisions. We maintain that the second sentence is an accurate summary of the limitations of the current body of evidence, and we note that a very similar sentence is used in the Abstract.
Reviewer 2 Report
I thank the authors for considering my comments and for their respective edits as well as for their counter-arguments. The authors have addressed efficiently my points A and B. However, regarding my points C (e.g. rendomization) and D. (intensity), I would ask the authors to consider a respective statement in the discussion section. Specifically: in my view the issue of randomization may prevent both from implementing long-term interventions and from bringing such studies to the fore-front. My question "randomize on what?" was not answered. The authors have acknowledged the difficulty of implementing long term interventions and suggesting/imposing a further abstract rule (randomize on what?) prevents from the advancement of the field and most importantly from the translation of research to practice. In my view, as long as there are no pre-intervention differences between the experimental and the control group in the target variables, then a researcher can have valid results and conclusions.
The same argumentation refers to the suggestion to equate intensity. Suggesting that prospective studies should equate intensity may be both out of context and also would restrict respective research unecessarily. For example, an individual is walking up a hill and from time to time s/he stops to enjoy the view. Some times s/he walks faster when the path is smooth and sometimes s/he walks slower when the path is stiff. Is there any meaning to try to equate this intensity in a study comparing outdoor with indoor exercise?
I respect the authors' counterarguments and understand that their study was not designed to address such issues. However, I would like to see in the discussion respective statements regarding the possible problems of these suggestions.
Author Response
While we appreciate the reviewer's perspective, below we maintain and elaborate on our views regarding randomization and exercise intensity. As a result, we do not feel that additional statements in the Discussion would be appropriate.
Reviewer Follow-Up C: In my view the issue of randomization may prevent both from implementing long-term interventions and from bringing such studies to the fore-front. My question "randomize on what?" was not answered. The authors have acknowledged the difficulty of implementing long term interventions and suggesting/imposing a further abstract rule (randomize on what?) prevents from the advancement of the field and most importantly from the translation of research to practice. In my view, as long as there are no pre-intervention differences between the experimental and the control group in the target variables, then a researcher can have valid results and conclusions.
Author Follow-Up C: We reiterate that we see randomization as a necessary tool in exercise interventions to mitigate bias and support the validity of the evidence.
In experimental studies, randomization—including simple randomization or restricted randomization—is crucial to help prevent selection bias and equalize potential confounders—both known and unknown—between treatment groups [1,2]. As such, it is well established as the primary method to mitigate “pre-intervention differences between the experimental and control group” in known and/or unknown variables that may affect the outcome(s) [1]. Allocating participants to groups without randomization (i.e., a chance, unpredictable process) and instead with any non-random, deterministic method (e.g., alternation, date of birth, use of pre-existing groups) is much more likely to introduce selection bias and imbalances in known or unknown confounders [1].
In response to the question, “randomize on what?”, we understand that the reviewer is referring to restricted randomization and giving examples of various participant characteristics with which to perform stratification. The ideal goal of randomization is to distribute these participant characteristics approximately evenly across both groups [1,2]. Researchers need to make their best judgement within the context of their intervention and study population to decide 1) whether this is most likely to be achieved via simple randomization or, especially with smaller samples, via restricted randomization, and 2) if using stratified randomization, which potentially confounding participant characteristic(s) is/are most important to use (e.g., sex) [1]. Regardless of what researchers decide, our main point is that, to reduce bias and confounding, they should perform some type of randomization (appropriate to their study) and report the details.
In our review, three of the included studies failed to mention any randomization, and most included studies failed to report the randomization type and/or mechanism (e.g., simple randomization by computer-generated random numbers), stating only that the study was randomized. Details on the randomization process (i.e., type, mechanism, allocation concealment) are important for readers to evaluate whether the allocation sequence was genuinely random and generated and implemented without bias [1,2]; trials with inadequate or unclear allocation sequence generation and allocation concealment tend to overestimate intervention efficacy [3].
Reviewer Follow-Up D: The same argumentation refers to the suggestion to equate intensity. Suggesting that prospective studies should equate intensity may be both out of context and also would restrict respective research unecessarily. For example, an individual is walking up a hill and from time to time s/he stops to enjoy the view. Some times s/he walks faster when the path is smooth and sometimes s/he walks slower when the path is stiff. Is there any meaning to try to equate this intensity in a study comparing outdoor with indoor exercise?
Author Follow-Up D: Thank you for this comment. We reiterate that, in a study comparing outdoor versus indoor exercise, the purpose of equating the prescribed intensity—as much as possible within the study context—is to isolate the effects of the independent variable (the physical exercise environment) from the potentially confounding effects of exercise intensity. We emphasize that the priority is that both groups are prescribed the same or sufficiently similar exercise parameters, including intensity, and that these details are clearly reported. For example, participants may be prescribed a specific target heart rate, or they may be instructed to choose their own pace and stop whenever desired; as long as the instruction is the same across both groups, effects on outcomes can be attributed to the environment. If there are differences between groups in the intensity prescribed by researchers or in measures taken by researchers to promote adherence to the prescribed intensity, or if these details are not reported, it is possible that these factors might confound the effects of the exercise environment.
We note that it is possible that, regardless of the prescribed intensity, the environment may cause participants to exercise at different actual intensities (e.g., as mentioned in the reviewer’s example, participants may exercise at a lower overall intensity outdoors versus indoors because they stop periodically to enjoy the view). The priority remains that both groups were prescribed the same—or as similar as possible—exercise protocols, including intensity, such that any differences in outcomes—or in the actual intensity achieved—can be attributed to the environment, rather than non-environment confounders within the exercise protocol.
References:
- Moher, D.; Hopewell, S.; Schulz, K.F.; Montori, V.; Gøtzsche, P.C.; Devereaux, P.J.; Elbourne, D.; Egger, M.; Altman, D.G. CONSORT 2010 Explanation and Elaboration: Updated Guidelines for Reporting Parallel Group Randomised Trials. BMJ 2010, 340, 869–887, doi:10.1136/BMJ.C869.
- Sterne, J.A.C.; Savović, J.; Page, M.J.; Elbers, R.G.; Blencowe, N.S.; Boutron, I.; Cates, C.J.; Cheng, H.Y.; Corbett, M.S.; Eldridge, S.M.; et al. RoB 2: A Revised Tool for Assessing Risk of Bias in Randomised Trials. BMJ 2019, 366, doi:10.1136/bmj.l4898.
- Page, M.J.; Higgins, J.P.T.; Clayton, G.; Sterne, J.A.C.; Hróbjartsson, A.; Savović, J. Empirical Evidence of Study Design Biases in Randomized Trials: Systematic Review of Meta-Epidemiological Studies. PLoS One 2016, 11, doi:10.1371/JOURNAL.PONE.0159267.